# Global Buckling Resistance of Cold-Formed Steel Beams with Omega-Shaped Sections

Rita Peres [1,*], José Carvalho [1], Jean Antonio Emerick [2], Luís Macedo [3], José Luiz Rangel Paes [1,2] and José Miguel Castro [1]

1. CONSTRUCT, Faculdade de Engenharia da Universidade do Porto, 4200-465 Porto, Portugal; jose.fontao@fe.up.pt (J.C.); jlrangel@ufv.br (J.L.R.P.); miguel.castro@fe.up.pt (J.M.C.)
2. Departamento de Engenharia Civil, Universidade Federal de Viçosa, Viçosa 36570-900, Brazil; jean.emerick@ufv.br
3. Departamento de Engenharia Mecânica, Instituto Superior de Engenharia do Porto, 4249-015 Porto, Portugal; lam@isep.ipp.pt
* Correspondence: ritaperes@fe.up.pt

**Abstract:** The absence of analytical expressions in current codes for evaluating the critical moment for lateral–torsional buckling of cold-formed beams with omega-shaped sections presents a fundamental challenge when assessing their resistance to global buckling. In response to this challenge, a comparative study was conducted to explore various approaches for calculating the critical moment. This involved both analytical and numerical analyses, using different methods available in codes and computational tools. The analytical analysis followed the Effective Width Method, employing the expression proposed in ENV 1993-1-1:1992, which is commonly used for evaluating the critical lateral–torsional moment of hot-rolled profiles. Numerical analyses were then performed using the ABAQUS v6.13, GBTUL v2.0, and CUFSM v5.05 software packages. The ABAQUS model, validated with results obtained from an experimental campaign, serves as the reference model. Upon assessing the bending moment resistances according to European, Brazilian, and American standards, consistency was found among these standards. However, it became evident that using the analytical expression proposed for hot-rolled profiles is inadequate for evaluating the critical lateral–torsional moment of CFS omega-shaped profiles. Conversely, the agreement between the ABAQUS, GBTUL, and CUFSM results suggests their utility as reliable tools for estimating the elastic critical lateral–torsional buckling moment.

**Keywords:** CFS omega-shaped beams behavior; global buckling of CFS; critical lateral–torsional buckling moment; experimental analysis; numerical analysis; ABAQUS; GBTUL; CUFSM

## 1. Introduction

In steel constructions, the use of Cold-Formed Steel (CFS) profiles in beams is a well-established and widely adopted practice. CFS profiles offer numerous advantages, including a high strength-to-weight ratio, versatility in manufacturing across a diverse range of cross-sectional shapes, reduced material consumption, and straightforward on-site installation. These characteristics collectively make CFS a popular and sustainable choice for various construction applications.

CFS beams feature a variety of cross-sectional shapes, with the most common being the channel section, Z section, and hat section. The latter is particularly noteworthy due to its ability to achieve larger clear spans compared to other sections. Additionally, by bending the flanges and the depth of the section, it is possible to significantly enhance its stiffness. A specific type of the hat section is the omega-shaped section, which incorporates multiple bends, enabling even larger structural spans and reduced vertical displacements.

CFS members have been extensively investigated due to their inherent slenderness, featuring thin-walled open or closed cross-sections. This characteristic makes them highly

susceptible to various instability phenomena, including local (L), distortional (D), shear (S), and global (G) flexural or flexural–torsional buckling, as well as couplings between them (e.g., L–D, L–G, D–G, or L–D–G mode coupling/interaction). These instability phenomena significantly impact the overall structural response and ultimate strength of CFS members, justifying their inclusion in CFS specifications. Consequently, CFS members have played a crucial role in the development and widespread adoption of the Direct Strength Method (DSM) [1].

Traditionally, thin-walled steel members were designed against local buckling using the Effective Width Method (EWM), an approach still embedded in current versions of certain codes, such as Eurocode 3 [2,3]. However, the evolution of increasingly complex cross-sectional shapes, coupled with the recognition of distortional buckling as a potential failure mode for cold-formed thin-walled members with lipped cross-sections, paved the way for the development of the DSM. In response to unsuccessful attempts to efficiently predict distortional failures using methods based on an "effective cross-section", a more rational approach was sought. This research effort led to the standardization of the DSM, initially in North America [4], and almost simultaneously included in the Australian/New Zealand Standard [5]. A few years later, the DSM approach was also incorporated into the Brazilian standards for CFS structures [6].

Numerous researchers have made substantial contributions to the development of the DSM. More recently, Martins et al. [7] reported on the buckling and failure of cold-formed steel simply supported beams under uniform bending, considering three cross-section shapes: (i) lipped channels bending about the major axis, (ii) zed-sections under tilt bending, causing uniform flange compression, and (iii) hat-sections subjected to either major-axis or minor-axis bending. This study revealed that the current DSM design curve fails to adequately predict the failure moments of some analyzed beams. To address this, the authors proposed novel DSM strength curves, offering better predictions of all available numerical failure moments. In a subsequent study, Martins et al. [8] introduced two DSM design approaches to account for local–distortional (L–D) buckling modes interactions for cold-formed hat-sections, demonstrating that the DSM is current suitable for evaluating the buckling modes of this type of section.

Additionally, several recent studies have been conducted to experimentally assess instability modes in steel elements subjected to bending loads. Nguyen et al. [9] and Piotrowski et al. [10] numerically investigated the elastic critical lateral–torsional buckling of steel beams under bending loads using the finite element software ABAQUS. Dib, Ramos, and Vieira [11] analyzed the structural behavior of cold-formed steel hat-section beams under non-uniform bending, using GBTUL and ABAQUS. De'nan et al. [12,13] numerically evaluated the buckling behavior of hat-shaped sections with holes along the section. Ma, Rasmussen, and Zhang [14] and Dar et al. [15] conducted four-point bending tests on cold-formed steel beams with different innovative cross-sectional shapes to assess their potential instability modes. Furthermore, Tikate and Sonar [16] presented an extensive review of recent advancements in the design and analysis of cold-formed steel (CFS) under bending, highlighting the benefits of including intermediate reinforcements at the flange/web junction.

The buckling behavior of cold-formed hat-shaped sections under bending has been experimentally investigated by Manikandan et al. [17] and Aktepe et al. [18]. However, despite the recent surge in popularity of CFS omega-shaped beams, the buckling behavior of these hat-section types, characterized by additional stiffeners along the section, has not yet been reported in any study, revealing a significant gap in the available scientific knowledge. The main goal of this study is to contribute to filling this gap, aiming to enhance the understanding of the structural behavior of CFS omega-shaped beams. Specifically, the focus is on evaluating the global buckling phenomenon as it represents a prominent failure mode in thin-walled elements with large spans.

The primary objective of this study is to investigate the behavior of omega-shaped sections, specifically focusing on the global buckling phenomena. The assessment of bending

moment resistance to global buckling is made using the European [2], Brazilian [6], and North American [4] standards.

In the context of the European [2] provisions, the evaluation of global moment resistance relies on the effective elastic section modulus, determined using the method of effective widths, and a buckling resistance factor. This factor is a function of the normalized slenderness, which, in turn, depends on the critical elastic lateral–torsional buckling moment. Notably, the European code lacks an equation to estimate the critical moment for CFS members. As a common practice, engineers often resort to using the expression from ENV 1993-1-1:1992 [19], recommended for evaluating the critical lateral–torsional moment of hot-rolled profiles.

The Brazilian [6] and North American [4] standards adopt the DSM. This method involves calculating the critical elastic local, distortional, and global buckling moments through an elastic buckling analysis, typically conducted using CUFSM [20–22]. The characteristic resistant bending moment is then determined as the smallest value calculated for global, local, and distortional buckling. If the profile does not exhibit local and distortional buckling modes in the analysis, the respective resistance corresponds to global buckling resistance, which is, again, directly related to the critical lateral–torsional moment.

Therefore, assessing the bending moment resistance to global buckling in CFS omega-shaped profiles depends on evaluating their critical moments. To achieve this, we conducted a comprehensive comparative study involving numerical analyses, using different computational tools, namely ABAQUS [23], GBTUL [24], and CUFSM v5.05 [15], and the analytical expression proposed in ENV 1993-1-1:1992 [19]. The ABAQUS model served as our reference, validated against results obtained from an experimental campaign. Additionally, this campaign enabled us to compare the profiles' sectional resistance with the values derived from the European [2], Brazilian [6], and North American [4] standards.

## 2. Normative Procedures for Assessing Buckling Safety

### 2.1. EN 1993-1-3:2006 (EC3)

The European standard EN 1993-1-3:2006 [2] recommends the Effective Width Method (EWM) for the assessment of buckling in cold-formed profiles for global, local, and distortional modes. This method is often considered highly conceptual and less practical, particularly for profiles characterized by numerous bends.

The general procedure for buckling verification according to EC3 [2] involves the following steps: (1) calculating the critical elastic buckling stress and identifying the susceptible buckling modes for various half-wavelengths up to the real length of the bar; (2) determining the effective width of the curved zones of the cross-section based on the minimum local buckling stress; (3) calculating the reduced thickness of end stiffeners, intermediate stiffeners, or other parts of the cross-section subjected to distortional buckling based on the minimum distortional buckling stress; and (4) calculating the global buckling resistance for the real length of the bar based on the effective cross-sectional area.

$$M_{b,Rd} = \chi_{LT} W_{eff,y} \frac{f_y}{\gamma_{M1}} \tag{1}$$

where $W_{eff,y}$ is the effective elastic section modulus, calculated according to the EWM for local and distortional buckling, and $\gamma_{M1}$ is the reduction coefficient for the steel profile strength, following EN 1993-1-1:2009 [25].

The buckling resistance coefficient, $\chi_{LT}$, is a function of the normalized slenderness, $\underline{\lambda}_{FLT}$:

$$\underline{\lambda}_{FLT} = \sqrt{\frac{W_y f_y}{M_{cr}}} \tag{2}$$

The normalized slenderness, in turn, depends on the value of the critical elastic lateral–torsional buckling moment. In the context of hot-rolled profiles with I or H sections, it is

common practice to adopt the equation for calculating the critical lateral–torsional moment presented in the European pre-standard ENV 1993-1-1:1992 [19].

$$M_{cr,LT} = C_1 \frac{\pi^2 E I_z}{L^2} \left\{ \left[ \left(\frac{k}{k_w}\right) \frac{I_w}{I_z} + \frac{(kL)^2 G I_t}{\pi^2 E I_z} + (C_2 z_g - C_3 z_j)^2 \right]^{0.5} - (C_2 z_g - C_3 z_j) \right\} \quad (3)$$

where $C_1$, $C_2$, and $C_3$ are factors depending on the loading and boundary conditions and $I_z$, $I_w$, and $I_t$, are, respectively, the moment of inertia about the z-z axis, the warping inertia of the section, and the torsional inertia. The coefficients $k$ and $k_w$ are associated with rotational restraints at supports, while $z_g$ is the distance from the point of load application to the shear center, and $z_j$ is the factor representing profile asymmetry. In Equation (3), boundary conditions are altered by varying the value of the coefficient $k$. To account for section warping, $k_w$ is considered to be 1.0, and if the rotation is desired to be fixed, $k_w$ is adopted as 0.5.

*2.2. ABNT NBR 14762:2010*

Annex C of ABNT NBR 14762:2010 [6] introduced a method for the design of cold-formed steel profiles subjected to simple bending, known as the Direct Strength Method (DSM). This method serves as an alternative to the Effective Width Method (EWM), and the prescriptions included in Annex C of this standard can be applied to calculate the resistant bending moment of the profile ($M_{Rd}$).

The critical elastic local buckling moment ($M_l$), distortional buckling moment ($M_{dist}$), and global buckling moment ($M_e$) should be calculated through an elastic buckling analysis, typically performed using computational tools, such as CUFSM [20], ABAQUS [23], and GBTUL [24]. If the profile does not exhibit one of the three buckling modes in the analysis, there is no need to consider the corresponding resistance.

The characteristic resistant bending moment ($M_{Rk}$) is taken as the smallest value calculated for global, local, and distortional buckling ($M_{Re}$, $M_{Rl}$, $M_{Rdist}$), respectively. The calculated resistant bending moment ($M_{Rd}$) is given by $\frac{M_{Rk}}{\gamma}$, where $\gamma$ is a partial safety factor that equals to 1.10. The equations for determining the value of the resistant bending moment to buckling are presented below.

2.2.1. Global Buckling

$$M_{Re} = W f_y \quad (for \ \lambda_0 \leq 0.6) \quad (4)$$

$$M_{Re} = 1.1(1 - 0.278\lambda_0^2) W f_y \quad (for \ 0.6 < \lambda_0 \leq 1.336) \quad (5)$$

$$M_{Re} = \frac{W f_y}{\lambda_0^2} \quad (for \ \lambda_0 \geq 1.336) \quad (6)$$

with

$$\lambda_0 = \sqrt{\frac{W \cdot f_y}{M_e}} \quad (7)$$

where $M_{Re}$ is the resistant bending moment to global buckling, $\lambda_0$ is the reduced slenderness ratio related to the global buckling mode, $W$ is the elastic section modulus about the bending axis, $f_y$ is the yield strength of the steel, and $M_e$ is the critical elastic bending moment for global buckling.

To determine the design resistance moment for global buckling, it is necessary to consider the distribution of bending moments along the laterally restrained segment. To this end, one should multiply the resistant bending moment to global buckling ($M_{Re}$) by the modification factor for non-uniform bending moment diagram ($C_b$), given by:

$$C_b = \frac{12.5 M_{max}}{2.5 M_{max} + 3 M_A + 4 M_B + 3 M_C} R_m \leq 3.0 \quad (8)$$

where $M_{max}$ is the value of the maximum bending moment acting on the unbraced length; $M_A$ is the value of the bending moment acting at a quarter of the unbraced length, measured from the left end; $M_B$ is the absolute value of the bending moment acting at the central section of the unbraced length; $M_C$ is the absolute value of the bending moment acting at three-quarters of the unbraced length, measured from the left end; and $R_m$ is a parameter that takes into account the asymmetry of the cross-sectional shape, equal to $0.5 + 2(I_{yc}/I_y)$ for sections with one symmetry axis, subject to bending about a non-symmetry axis and experiencing reverse curvature, and is equal to 1.0 in all other cases, where $I_{yc}$ is the moment of inertia of the compressed flange about the symmetry axis.

To obtain the exact value of $M_e$, since the numerical analysis of elastic stability provides an approximate result, it is recommended to use the following equation for cold-formed profiles with monosymmetric sections, derived for loading applied at the position of the torsion center and subject to bending about an axis perpendicular to the symmetry axis (Annex E—ABNT NBR 14762:2010 [6]):

$$M_e = \frac{C_s N_{ex}}{C_m} \left[ j + C_s \sqrt{j^2 + r_0^2 \left( \frac{N_{ez}}{N_{ex}} \right)} \right] \tag{9}$$

where $C_s = +1$ if the bending moment causes compression in the section part with a negative x-coordinate, i.e., on the same side as the torsion center; $C_s = -1$ if the moment causes tension in the section part with a negative x-coordinate, i.e., on the same side as the torsion center; $C_m$ is the parameter depending on loading conditions; $r_0$ is the polar radius of gyration of the gross section about the torsion center; and j pertains to the factor representing the mono-symmetry of the profile. $N_{ex}$, $N_{ey}$, and $N_{ez}$ correspond to the axial forces of elastic global buckling due to bending about the x-x major axis, y-y minor axis, and z-z torsional axis, respectively:

$$N_{ex} = \frac{\pi E I_x}{(K_x L_x)^2} \quad N_{ey} = \frac{\pi E I_y}{(K_y L_y)^2} \quad N_{ez} = \frac{1}{r_0^2} \left[ \frac{\pi^2 E C_w}{(K_z L_z)^2} + GJ \right] \tag{10}$$

In these expressions above, $C_w$ is the warping constant of the section; $E$ is the longitudinal modulus of elasticity; $G$ is the transverse modulus of elasticity; $J$ is the torsional constant of the section; $I_x$ is the moment of inertia about the x-x major axis; $I_y$ is the moment of inertia about the y-y minor axis; $K_x L_x$ is the effective length of global buckling due to bending about the x-x major axis; $K_y L_y$ is the effective length of global buckling due to bending about the y-y minor axis; and $K_z L_z$ is the effective length of global buckling due to torsion about the z-z torsional axis.

2.2.2. Local Buckling

$$M_{Rl} = M_{Re} \quad (for\ \lambda_l \leq 0.776) \tag{11}$$

$$M_{Rl} = \left( 1 - \frac{0.15}{\lambda_l^{0.8}} \right) \frac{M_{Re}}{\lambda_l^{0.8}} \quad (for\ \lambda_l > 0.776) \tag{12}$$

with

$$\lambda_l = \sqrt{\frac{W \cdot f_y}{M_l}} \tag{13}$$

where $M_{Rl}$ is the resistant bending moment to local buckling; $\lambda_l$ is the reduced slenderness ratio related to the local buckling mode; and $M_l$ is the critical elastic bending moment for local buckling.

2.2.3. Distortional Buckling

$$M_{Rdist} = W f_y \quad (for\ \lambda_{dist} \leq 0.673) \tag{14}$$

$$M_{Rl} = \left(1 - \frac{0.22}{\lambda_{dist}}\right) \frac{W f_y}{\lambda_{dist}} \quad (for\ \lambda_{dist} > 0.673) \tag{15}$$

with

$$\lambda_{dist} = \sqrt{\frac{W \cdot f_y}{M_{dist}}} \tag{16}$$

where $M_{Rdist}$ is the resistant bending moment to distortional buckling; $\lambda_{dist}$ is the reduced slenderness ratio related to the distortional buckling mode; and $M_{dist}$ is the bending moment for distortional buckling in the elastic regime.

### 2.3. AISI S200:2016

Similar to the Brazilian standard ABNT NBR 14762:2010 [6], Annex F of the American specification AISI S200:2016 [4] employs the Direct Strength Method (DSM) to assess the safety of cold-formed steel profiles under simple bending. The formulation presented in [4] closely resembles that in [6], except for the treatment of the inelastic reserve of strength. In the American specification, this reserve is linked to global, local, and distortional buckling. In contrast, both the European and Brazilian specifications associate the inelastic reserve solely with the local buckling mode.

## 3. Experimental Campaign

One of the main objectives of the experimental campaign was to produce data to validate the ABAQUS finite element model. Two types of tests were conducted on a set of three omega-shaped section profiles: (i) tests involving the application of load in the downward direction, allowing for the estimation of positive sectional moment values, and (ii) tests involving the application of load in the downward direction on inverted profiles, aimed at evaluating negative sectional moment values. For each test, the key results to be retained include the load and displacement observed and failure modes.

### 3.1. Geometric and Material Properties

The profiles under investigation belong to a family of commercial profiles with an omega section, comprising six different heights, each with three or four distinct thicknesses. The selection aimed to include the smallest profile (depth of 70 mm), the largest one (depth of 300 mm), and an intermediate one (depth of 170 mm). In terms of thickness, a consistent logic was applied: the smallest thickness (1.5 mm) for the 70-mm profile, the largest thickness (2.5 mm) for the 300-mm profile, and an intermediate thickness (2.0 mm) for the 170-mm profile.

Table 1 provides the geometric characteristics of the selected profiles, while Table 2 presents the material properties evaluated based on tensile coupon tests.

**Table 1.** Geometric properties of the selected profiles.

| Profile Section | Weight (kg/m) | Height, $h$ (mm) | Width, $b$ (mm) | Length, $L$ (m) | Thickness | |
|---|---|---|---|---|---|---|
| | | | | | Effective, $t_{eff}$ (mm) | Nominal, $t_{nom}$ (mm) |
| 70 × 1.5 | 2.90 | 70 | 138 | 4.20 | 1.5 | 1.46 |
| 170 × 2.0 | 7.75 | 170 | 234 | 4.20 | 2.0 | 1.96 |
| 300 × 2.5 | 14.47 | 300 | 269 | 8.30 | 2.5 | 2.46 |

**Table 2.** Material properties.

| Profile Section | Yielding Stress, $f_y$ (MPa) | Ultimate Stress, $f_u$ (MPa) | Maximum Strain, $\varepsilon$ (%) |
|---|---|---|---|
| 70 × 1.5 | 340 | 455 | 27.5 |
| 170 × 2.0 | 330 | 453 | 31.6 |
| 300 × 2.5 | 374 | 476 | 29.0 |

The modulus of elasticity considered was 210 GPa, and the material density was 7850 kg/m$^3$.

### 3.2. Test Setup

The experimental tests followed the requirements outlined in Annex A.3 of EN 1993-1-3:2006 [2]. Consequently, the omega profiles 70 × 1.5 and 170 × 2.0 underwent testing using Layout 1, whilst the omega profiles 300 × 2.5 were subjected to testing according to Layout 2.

Layout 1 has a free span of 4.0 m, and the beam is simply supported at both ends. The profile is loaded at four points, as illustrated in Figure 1. The load transfer from the actuator is carried out through an IPE 100 beam supported by two metal plates with dimensions of 46 cm × 10 cm × 2 cm each. Each of these plates discharges onto two metal plates with dimensions of 15 cm × 10 cm × 1 cm, placed on neoprene bands, through rollers. LVDTs (extensometers) were positioned at mid-span and at the ends of the constant moment span, which has a length of 800 mm (corresponding to 20% of the total span between supports), as indicated in Figure 1a.

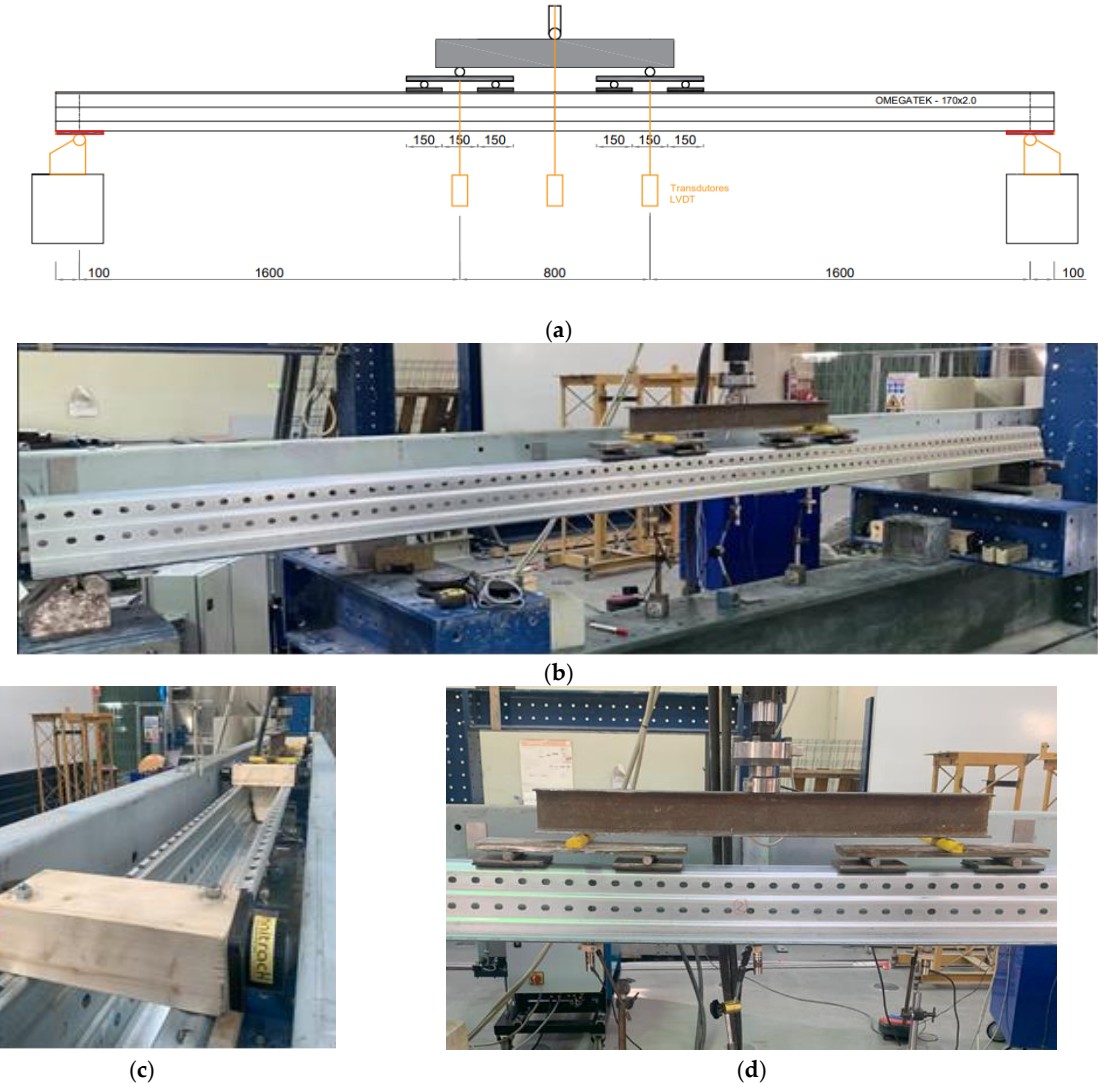

**Figure 1.** Test setup with a 4-m simply supported span: (**a**) schematic representation of Layout 1, (**b**) photograph of Layout 1 for applying a downward load, (**c**) photograph of Layout 1 for simulating upward load application with continuous lateral and punctual bracings, and (**d**) detail of the load scheme, depicting the actuator, loading beam and plates, and neoprene bands located between the load plates and the profile.

Layout 2 has a free span of 8.0 m, and the beam is also simply supported at both ends. Loading is applied at six points. The load transfer from the actuator is facilitated through an IPE 250 profile, supporting two metal plates with dimensions of 60 cm × 10 cm × 2 cm each. Each plate discharges onto three plates with dimensions of 15 cm × 10 cm × 1 cm, positioned on neoprene bands, through rollers. LVDTs (extensometers) were positioned at mid-span and at the ends of the constant moment span, which has a length of 2200 mm (corresponding to 27.5% of the total span between supports), as indicated in Figure 2a. All the tests were carried out by displacement control at a constant speed of 0.04 mm/s.

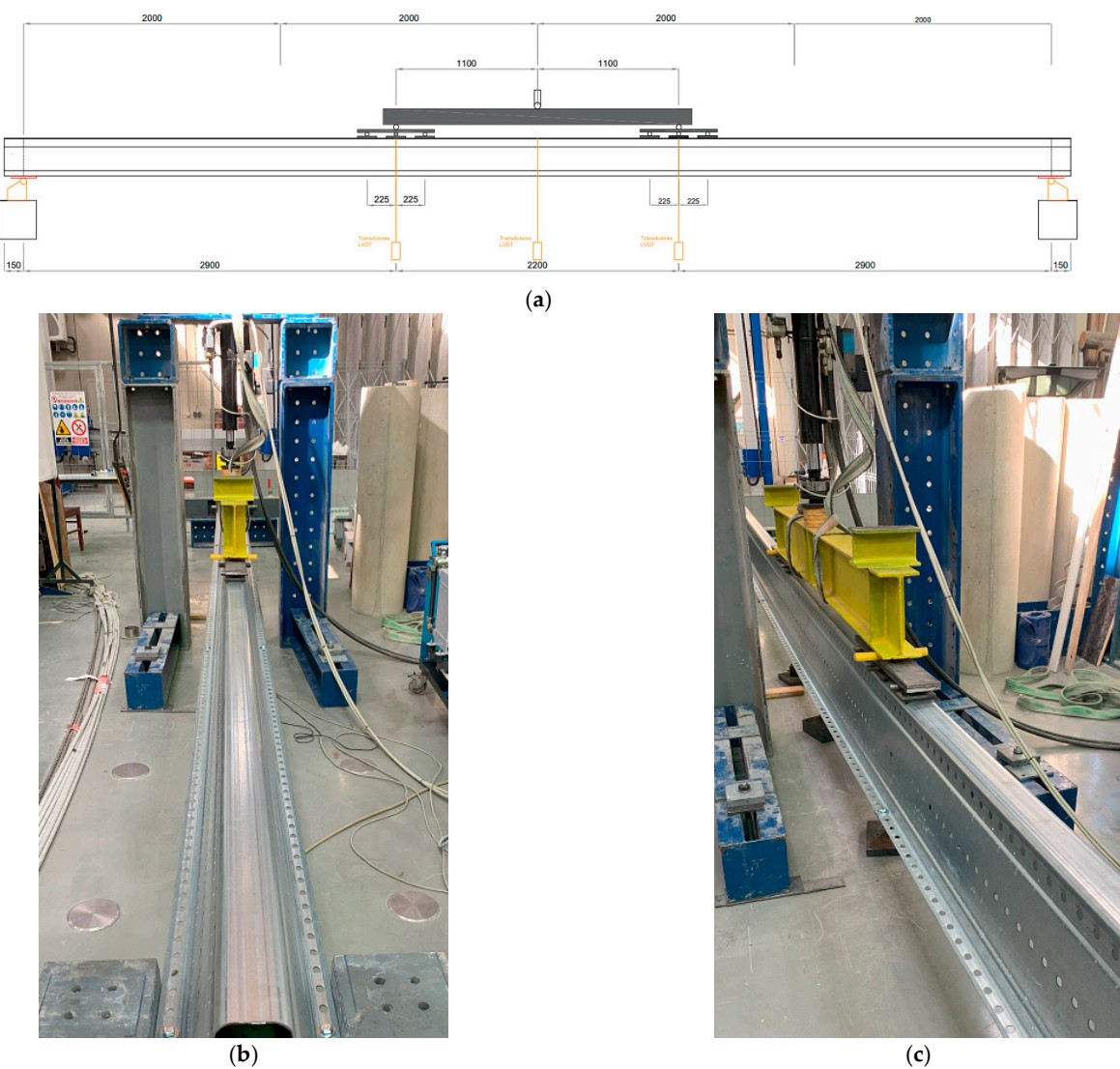

**Figure 2.** Test setup with an 8-m simply supported span: (**a**) schematic representation of Layout 2, (**b**) photograph of the test layout, and (**c**) detail of the loading scheme.

The profiles underwent testing in both downward and upward directions. To simulate upward loading, it was necessary to apply an ascending load on the continuous element, corresponding to loading the profiles in an inverted position. Lateral bracing, as shown in Figure 1c,d, was essential in the inverted position to prevent profile instability. In Figure 1c, continuous and punctual bracing can be observed. Punctual bracing was implemented in four and five locations for the 170 × 2.0 and 70 × 1.5 profiles, respectively. For the medium profile, punctual bracing was applied at the supports and immediately after the occurrence of the maximum moment as it was not possible to restrain at the location of the maximum span moment where the loading plates are located. The smaller profile had the same punctual bracing as the medium profile, with an additional brace at mid-span.

Due to the geometrical characteristics of Layout 2 (see Figure 2b,c), it was not feasible to continuously brace the profile in an inverted position to simulate loading in an upward direction. Consequently, tests involving only downward loads were conducted with this layout.

### 3.3. Experimental Results

Figures 3 and 4 illustrate the load–displacement curves recorded during the tests performed using Layout 1, applying both downward and upward loads, to profiles 70 × 1.5 and 170 × 2.0, respectively.

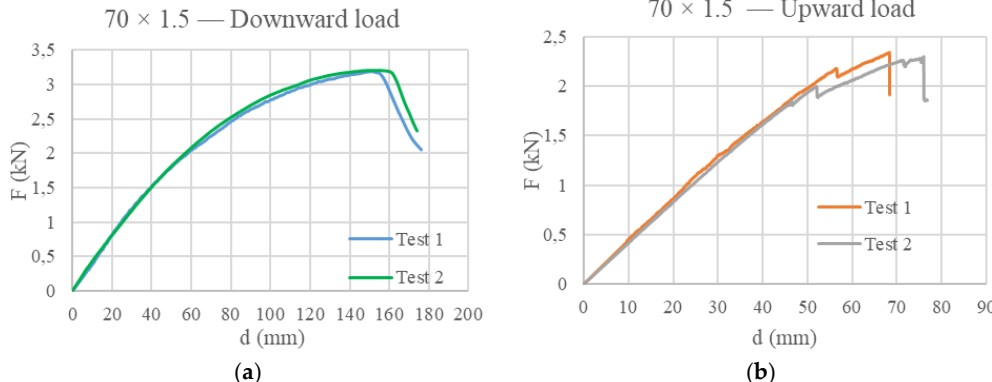

**Figure 3.** Experimental results of the tests carried out to the 70 × 1.5 profile under (**a**) a downward load and (**b**) an upward load.

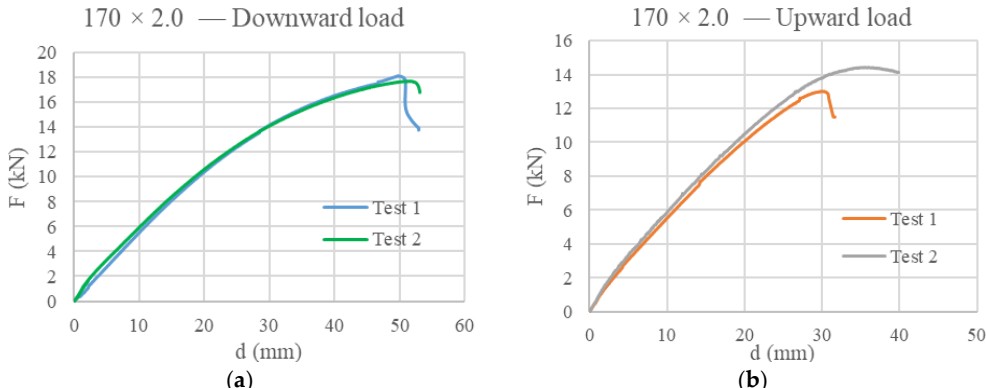

**Figure 4.** Experimental results of the tests carried out to the 170 × 2.0 profile under (**a**) a downward load and (**b**) an upward load.

Figure 5 illustrates the load–displacement curves recorded during the tests conducted on the 300 × 2.5 profile under a downward load.

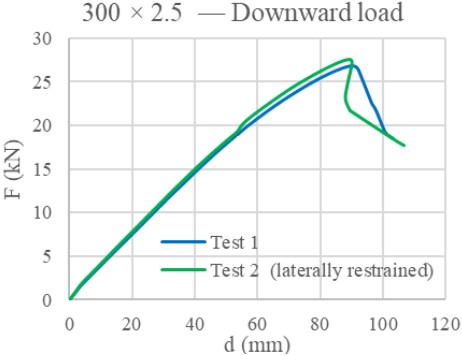

**Figure 5.** Experimental results of the tests carried out applying a downward load to a 300 × 2.5 profile.

The observations from Figures 3 and 4 indicate that, generally, the two experimental tests show consistent results in terms of observed loads and displacements when applying downward loads. However, when simulating upward loading, the results do not closely match due to the instability of the profiles when placed in the inverted position, necessitating continuous bracing. The drops in load observed in Figure 3b resulted from the $70 \times 1.5$ profile adjusting itself to the bracing system during loading. This phenomenon was not observed in the $170 \times 2.0$ profile (Figure 4b), likely because of its larger section and more effective bracing.

For the $300 \times 2.5$ profile, only downward loads were applied. The difference observed between the two test results is because, in Test 2, the profile was locally braced at the zero-moment span, resulting in a small increase in load compared to Test 1.

The next set of figures illustrates the failure modes observed during the experimental tests. Figures 6 and 7 depict the failure modes observed for both load application scenarios, for the $70 \times 1.5$ and $170 \times 2.0$ profiles. Meanwhile, Figure 8 illustrates the failure mode observed for the $300 \times 2.5$ profile under the application of a downward load.

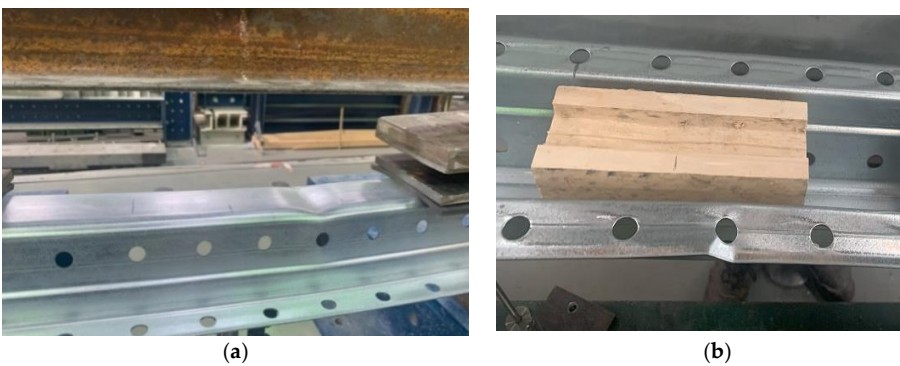

(**a**) (**b**)

**Figure 6.** Failure modes observed during the testing of the $70 \times 1.5$ profile under (**a**) a downward load and (**b**) an upward load.

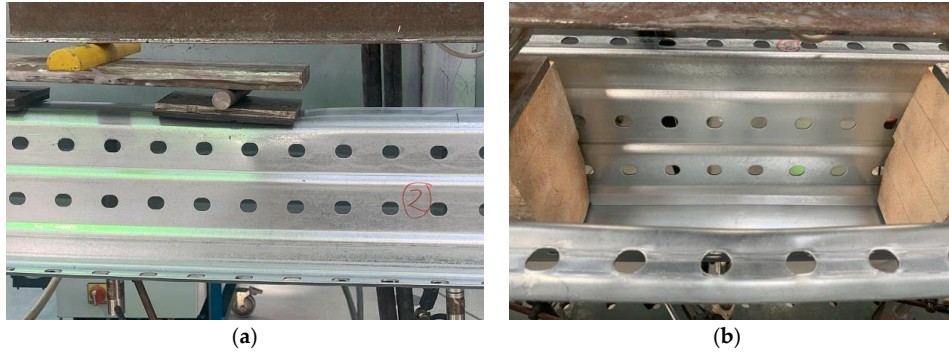

(**a**) (**b**)

**Figure 7.** Failure modes observed during the testing of the $170 \times 2.0$ profile under (**a**) a downward load and (**b**) an upward load.

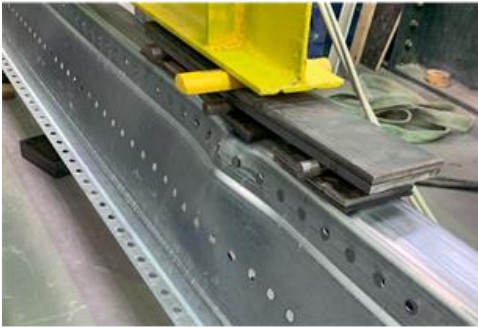

**Figure 8.** Failure modes observed during the testing of the $300 \times 2.5$ profile.

### 3.4. Comparison with Design Codes

The failure modes observed during the tests conducted on the $70 \times 1.5$, $170 \times 2.0$, and $300 \times 2.5$ profiles under downward loads are local failures, occurring near or in zones where the loads are applied, as illustrated in Figures 6a, 7a and 8. However, when upward loads are applied, the $70 \times 1.5$ and $170 \times 2.0$ profiles primarily exhibit distortional failure modes.

The maximum loads obtained during the experimental tests are compared to the anticipated loads when applying both the Effective Width Method (EWM) and the Direct Strength Method (DSM), as indicated, respectively, by the European [2], Brazilian [6] and North American [4] standards.

The Effective Width Method (EWM) involves calculating the effective properties of the cross-section, considering the reduction of resistance due to local and distortional buckling modes. The global resistant moment is computed using effective properties influenced by a buckling resistance factor, which, in turn, depends on the critical lateral–torsional moment value. On the other hand, the application of the Direct Strength Method (DSM) involves calculating the critical elastic local, distortional, and global buckling moments through an elastic buckling analysis, conducted using CUFSM. The characteristic resistant bending moment is determined as the smallest value calculated for global, local, and distortional buckling.

In the case of downward load application, the observed failure modes are mainly local, occurring near or in zones where the loads are applied. Therefore, for the purpose of sectional resistance evaluation and prediction of the maximum loads, the global buckling phenomena can be ignored. The comparison of the maximum recorded loads and expected loads, as illustrated in Table 3, confirms that both the EWM and DSM provide consistent results since the analytical expected loads closely align with the recorded experimental loads.

**Table 3.** Maximum recorded loads and expected loads according to different design codes.

| Profile Section | Recorded Loads (kN) | Expected Loads Using EWM [2] (kN) | Expected Loads Using DSM [4,6] (kN) |
|---|---|---|---|
| $70 \times 1.5$—Test 1 | 3.20 | | |
| $70 \times 1.5$—Test 2 | 3.21 | 3.12 | 3.36 |
| $170 \times 2.0$—Test 1 | 18.13 | | |
| $170 \times 2.0$—Test 2 | 17.68 | 17.15 | 17.75 |
| $300 \times 2.5$—Test 1 | 26.83 | | |
| $300 \times 2.5$—Test 2 | 27.51 | 28.90 | 32.22 |

Additionally, it was observed that the expected load values calculated with the EWM for the profiles $70 \times 1.5$ and $170 \times 2.0$ are slightly lower than the recorded load values, indicating that the European standard provides conservative estimates of bending resistance. DSM yields slightly higher expected loads compared to the EWM, although the maximum observed difference between these methods is 10%, and this pertains to the $300 \times 2.5$ profiles.

## 4. Numerical Models

### 4.1. Developed Models

Three numerical models were developed to evaluate the critical elastic moment for lateral–torsional buckling of the omega-shaped sections tested during the experimental campaign. The selected programs for conducting the analyses were ABAQUS, a software employing the Finite Element Method, GBTUL, which implements the Generalized Beam Theory for analyzing buckling phenomena in bending, torsion, and overall buckling, and CUFSM (Constrained and Unconstrained Finite Strip Method), a software for evaluating buckling modes based on the Finite Strip Method (FSM). The validation of the ABAQUS model was conducted using the results obtained from the experimental campaign because,

being a finite element software, it is considered to be the most accurate representation of the real behavior of the tested profiles.

### 4.2. Development and Validation of the ABAQUS Model

The omega-shape profiles were simulated using a shell model, using the SR8 reduced integration finite element type available in ABAQUS, representing the mid-surface of the profile's geometry. The length of the profile was generated by applying the extrusion command. The holes that exist along the profiles, whether circular or oval-shaped, were also included in the model. In the material characterization, yield, and ultimate stresses, as well as the ultimate strain of the material, were considered. The simulation of the material's nonlinear behavior used stress–strain curves defined by the Ramberg-Osgood model [26]. All the other components that were used in the model were simulated with solid parts made to replicate the laboratory setup (Figure 9). The material for these parts was assumed to be linear elastic. The modeling of the loading and boundary conditions for the two layouts used in the experimental campaign is detailed in [27].

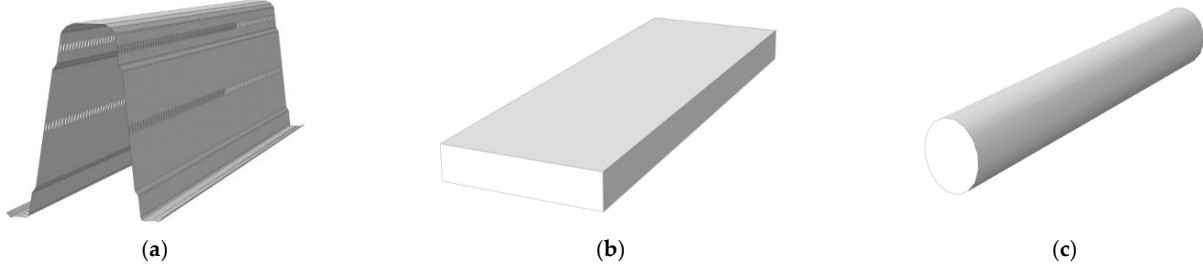

(**a**)        (**b**)        (**c**)

**Figure 9.** Parts used in the models: (**a**) profile section 300 × 2.5, (**b**) contact plate with the profile, and (**c**) support cylinder.

The contact between the neoprene bands that were placed between the omega profiles and the other test setup parts was modeled using General Contact with a friction coefficient of 0.35, representing the friction between steel and rubber. Initially, the friction coefficient specified in the American standard ANSI AISC 360-22 [28], set at 0.30, was used, which represents the friction coefficient for steel surfaces free of oil, grease, or painting. Subsequently, the value established in the Brazilian standard ABNT NBR 8800:2008 [29] for the same steel surfaces, set at 0.35, was adopted. The use of this latter value resulted in greater consistency between the numerical results and those obtained experimentally.

The loading was applied to accurately replicate the experimental tests. To simulate the hydraulic actuator used in the experiments, the load was applied at a single point. Subsequently, it was evenly distributed onto the four loading plates using a coupling-structural-uniform restriction (see Figure 10). The contact plates were steel plates with the dimensions of 15 cm × 10 cm × 1 cm, as described in Section 3.2 (Test setup).

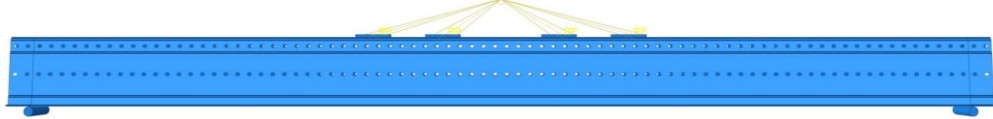

**Figure 10.** Modeling of the loading scheme for the 170 × 2.0 profile (Layout 1).

The boundary conditions vary depending on the profile size. For larger sections, it is necessary to model the cylinders to account for local deformation occurring at the supports. The cylinders modeled to simulate the supports are steel elements with a 2 cm radius and 30 cm length. In the case of smaller profiles, restraints are applied at four points, corresponding to the contacts with the supports, where vertical displacements (UY) and lateral displacements (UX) are fixed (Figure 11a). In contrast, the larger profiles interact with cylinders through friction, and the cylinders are fixed against all displacements (UX, UY, UZ), as illustrated in Figure 11b. Additionally, in some tests, lateral displacements needed to

be restrained to address instability during upward loading. In such cases, lateral displacements (UX) were fixed at the same locations on the profiles as in the experimental tests. For the 70 × 1.5 profile, these locations are at the supports, immediately after the occurrence of the maximum moment (see Figure 3c), where punctual bracing was applied, and at the mid-span. In the case of the 170 × 2.0 profile, the locations are the same as for the smaller profile, except for the mid-span.

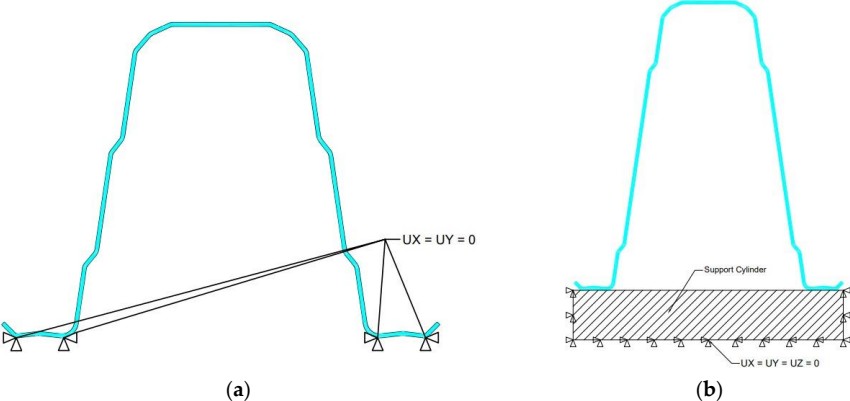

(a)                                     (b)

**Figure 11.** Boundary conditions: (**a**) small profiles such as 70 × 1.5 and 170 × 2.0 and (**b**) larger profiles such as 300 × 2.5.

Another noteworthy aspect is the meshing process. Given the hole pattern and bends present in the profile, a refined mesh is necessary. A mesh size of $5 \times 5$ mm$^2$ was chosen to ensure reasonable consistency throughout the generated mesh.

The experimental results were collected from various points of the profile. The loading data were obtained from the point of load application, while the displacements were measured at the points where the LVDTs (Linear Variable Differential Transformers) were located.

The validation of the models for the three analyzed profiles involved a comparison of the load and displacement values recorded during the tests with those obtained through numerical simulation. Additionally, the observed failure modes were compared to those obtained through numerical simulation. Figure 12 presents a comparison of the load and displacement values obtained experimentally and analytically for the 170 × 2.0 profile.

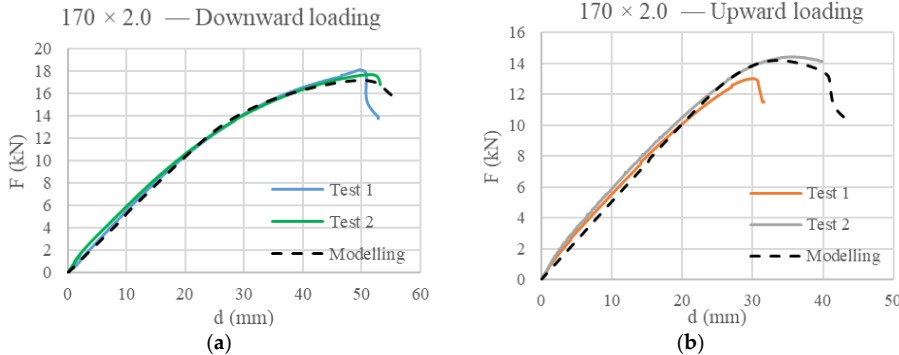

(a)                                     (b)

**Figure 12.** Comparing the results between numerical models and experimental tests on the 170 × 2.0 profile: (**a**) downward loading and (**b**) upward loading.

The analysis of Figure 12 yielded the following observations: in Figure 12a, the variation obtained for the force applied by the actuator was −4.07%, while the displacement was merely −1.45%, thus validating this model. Further validation is evidenced by the variation in Figure 12b, where the applied force exhibited a −3.4% change, while the displacement showed a −5.7% variation.

Figures 13 and 14 provide a comparison between the observed failure modes and those obtained from numerical simulation under downward and upward loading, respec-

tively. The numerical model demonstrates a high level of accuracy in capturing the experimental results.

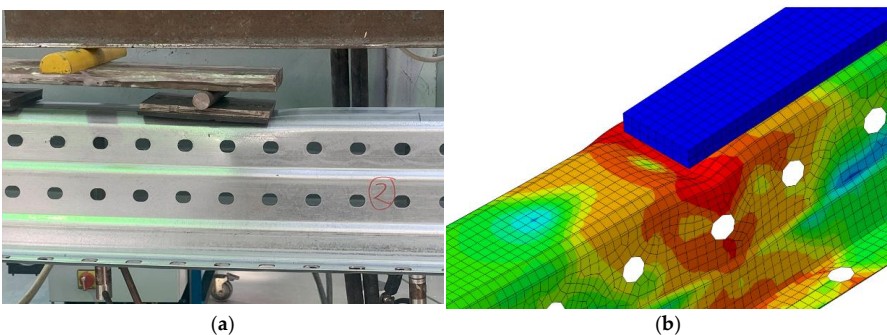

(**a**)           (**b**)

**Figure 13.** Failure modes of the 170 × 2.0 profile under downward loading: (**a**) failure in the experimental test and (**b**) failure in the ABAQUS model.

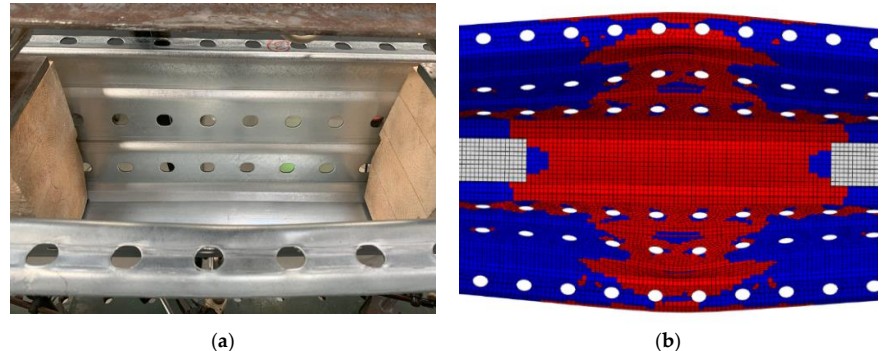

(**a**)           (**b**)

**Figure 14.** Failure modes of the 170 × 2.0 profile under uplift loading: (**a**) failure in the experimental test and (**b**) failure in the ABAQUS model.

Due to their capability to achieve larger clear spans, omega-shaped profiles are commonly utilized in industrial structures, often with cladding sheets attached. These sheets stabilize the profiles by being fixed to them. However, this beneficial effect was not considered both in the experimental campaign and in the numerical analysis.

### 4.3. GBTUL Model

The modeling process in GBTUL is straightforward due to its simple and clear interface. It involves the following steps:

Step 1: Definition of the geometry of the structure, including dimensions, shape, and material properties such as Young's modulus (210 GPa) and Poisson's ratio (0.3). All units are typically specified in the International System of Units (SI), with forces in kilonewtons (kN) and lengths in meters. Figure 15 shows the geometry input for the 170 × 2.0 profile.

Step 2: Validation of the buckling modes: before proceeding with the analysis, it is essential to validate the buckling modes of the structure. This involves confirming that all essential modes are present and that they make sense for the given case. For the three profiles under examination, all the buckling modes were considered in the analysis; however, the essential modes are the first five modes, which are the (1) axial mode; (2) major axis bending mode; (3) minor axis bending mode; (4) torsional mode; and (5) distortional mode.

Step 3: Application of loads and boundary conditions: within this step the software makes use of computational methods to analyze the structure under various loads and boundary conditions. To do so it is necessary to define the following parameters:

(1) The number of partitions/elements (mesh). The evaluation of the appropriate mesh density is crucial to accurately capture the behavior of the structure. This involves dividing the structure into smaller elements for numerical analysis. The mesh considered for the 170 × 2.0 profile with a length of 4.0 m was 20.

(2) Support conditions: all the tested profiles were simply supported on both sides.
(3) Define unit loads: the loads applied to the structure are point loads at the locations indicated in Layout 1 (see Figure 16) for the case of the $70 \times 1.5$ and $170 \times 2.0$ profiles and in Layout 2 for the case of the $300 \times 2.5$ profiles.
(4) Lengths to evaluate: the length of interest to simulate the experimental tests is 4.0 m.
(5) Number of eigenmodes to be calculated: the number of eigenmodes (buckling modes) depends on the complexity of the structure and the desired level of accuracy. For the $170 \times 2.0$ profiles, the number of eigenmodes calculated was 180.
(6) Observation of results: after performing the numerical analysis, observe the results to gain insights into the behavior of the structure under the applied loads and boundary conditions. The results obtained for the $170 \times 2.0$ profiles are shown in Figure 17.

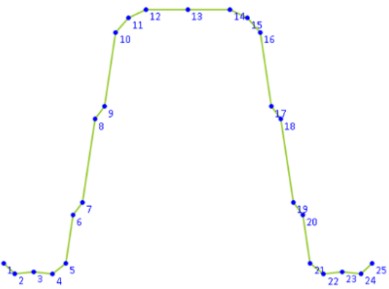

**Figure 15.** Geometry input for the $170 \times 2.0$ profile.

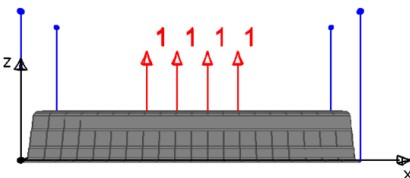

**Figure 16.** Load type and locations for the case of the $70 \times 1.5$ and $170 \times 2.0$ profiles.

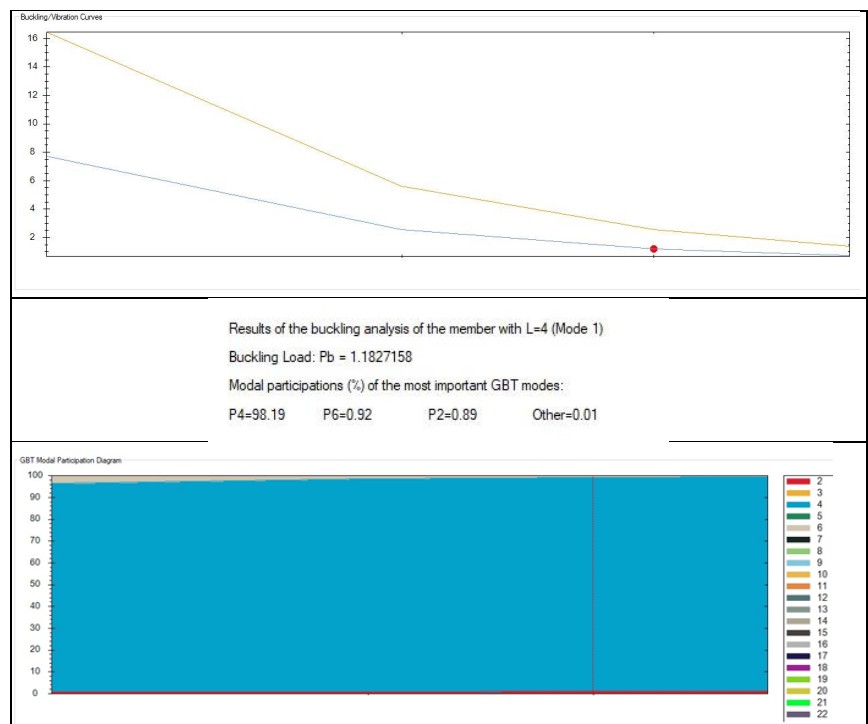

**Figure 17.** Buckling curve and GBT modal participation diagram obtained for the $170 \times 2.0$ profile.

The analysis of Figure 17 allows us to conclude that the buckling load is 1.18 kN, which corresponds to a critical elastic buckling moment of 3.77 kN.m, with a major contribution of the torsional mode (98%).

### 4.4. CUFSM Model

The CFS omega-shaped profiles were also numerically simulated using the software CUFSM v5.05 to determine the critical elastic buckling moments and possible omega section buckling modes. All the sections (70 × 1.5, 170 × 2.0, and 300 × 2.5) were analyzed for both upward and downward loading cases, but only the simulations performed in the 170 × 2.0 section are described.

In Figure 18, the geometric model of this section is presented, showing the stresses applied at the profile's critical section (in kN.m) for downward and upward loading, respectively. These stresses were applied assuming that the profile's upper flange reached the yield stress of the steel ($f_y$ = 330.0 MPa). The lines in blue represent compressive stresses, while the lines in red represent tensile stresses along the omega section.

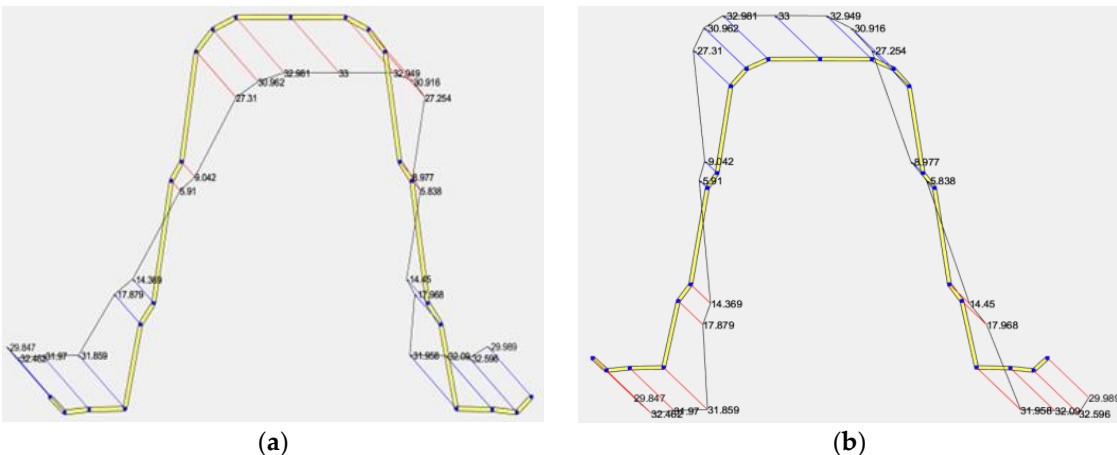

**Figure 18.** Critical section for the 170 × 2.0 profile under (**a**) downward loading and (**b**) upward loading.

To perform elastic buckling analysis in CUFSM, several assumptions were made: the signature curve method was chosen as the solution approach, and the beam support configurations were defined as simple–simple (S–S) to replicate the experimental setup's condition of a simply supported beam. Additionally, the analysis covered half-wavelengths ranging from 1.0 mm to 10,000 mm.

Figure 19 illustrates the signature curve of the 170 × 2.0 profile for upward loading. It is noteworthy that the curve's minimum points correspond to critical buckling modes. Moreover, the critical global buckling mode can be directly identified on the curve when the half-wavelength equals the beam's unbraced length, provided it is identified as the global mode through modal identification analysis. In Figure 19, the critical modes identified for the curve minima and the half-wavelength equal to the unbraced length (4000 mm) are highlighted, along with the corresponding critical elastic buckling moment, which is 1.46 kN.m.

It is important to note that CUFSM only allows the consideration of a uniform bending moment along the length of the profile. For other types of loading, such as non-uniform bending moment diagrams, a modification factor needs to be calculated [6], as outlined in Section 2.2. Consequently, the critical elastic buckling moments obtained with GBTUL and CUFSM do not align due to this limitation.

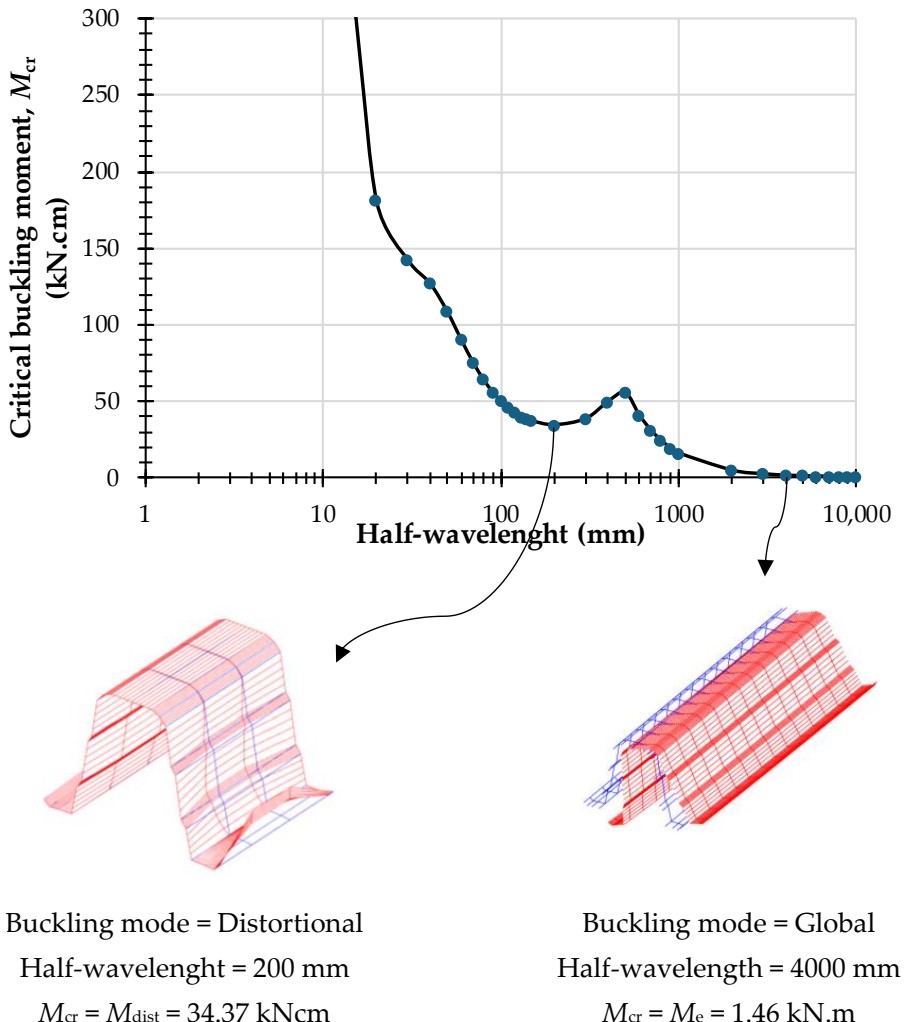

Buckling mode = Distortional

Half-wavelenght = 200 mm

$M_{cr} = M_{dist} = 34.37$ kNcm

Buckling mode = Global

Half-wavelength = 4000 mm

$M_{cr} = M_e = 1.46$ kN.m

**Figure 19.** CUFSM analysis results for the CFS 170 × 2.0 omega section under upward loading.

## 5. Global Buckling Bending Moment Resistance

In this section, the bending moment resistance to the global buckling of the selected Cold-Formed Steel (CFS) profiles is assessed according to the European [2], Brazilian [6], and North American [4] standards. Only negative moment scenarios have been considered. First, the critical elastic lateral–torsional buckling moment using Equation (3), provided by ENV 1993-1-1:1992 [14], is calculated, and then, through the numerical models developed in ABAQUS, GBTUL, and CUFSM. As mentioned earlier in this paper, Equation (3) is recommended for evaluating the critical lateral–torsional moment of hot-rolled profiles. Due to the absence of an analytical expression for CFS profiles in the European code, practitioners often resort to using this expression.

Regarding the numerical models, ABAQUS [16] is considered the reference model since it was validated with the results of the experimental campaign. It should also be noted that the critical moment obtained through the CUFSM [15] program was used to determine the resistance moment to global buckling by the Direct Strength Method (DSM), considering the modification factor $C_b$ from Equation (8). Tables 4–6 show the results associated with load cases $\psi = 1.0$, $\psi = 0.5$, and $\psi = 0.0$, as illustrated in Figure 20. The modification factor $C_b$ was utilized in correlation with the $\psi$ in these cases. For each profile, the results for two span lengths were analyzed.

**Table 4.** Results of the uplift critical moments for the 70 × 1.5 profile.

| Loading Cases | Profile 70 × 1.5 (*L* = 1.0 m) | | | | Profile 70 × 1.5 (*L* = 4.0 m) | | | |
|---|---|---|---|---|---|---|---|---|
| | $M_{cr}$ (Equation (3)) | $M_{cr}$ (GBTUL) | $M_e$ (CUFSM) | $M_{cr}$ (ABAQUS) | $M_{cr}$ (Equation (3)) | $M_{cr}$ (GBTUL) | $M_e$ (CUFSM) | $M_{cr}$ (ABAQUS) |
| | kN.m | kN.m | kN.m | kN.m | kN.m | kN.m | kN.m | kN.m |
| Ψ = 1.0 | 91.77 | 1.71 | 1.77 | 1.69 | 5.74 | 0.25 | 0.26 | 0.24 |
| Ψ = 0.5 | 120.21 | 2.26 | 2.22 | 2.20 | 7.80 | 0.33 | 0.33 | 0.32 |
| Ψ = 0.0 | 162.43 | 3.19 | 2.95 | 3.10 | 10.15 | 0.44 | 0.43 | 0.43 |

Note: L is the theoretical span of the beam.

**Table 5.** Results of the uplift critical moments for the 170 × 2.0 profile.

| Loading Cases | Profile 170 × 2.0 (*L* = 3.0 m) | | | | Profile 170 × 2.0 (*L* = 6.0 m) | | | |
|---|---|---|---|---|---|---|---|---|
| | $M_{cr}$ (Equation (3)) | $M_{cr}$ (GBTUL) | $M_e$ (CUFSM) | $M_{cr}$ (ABAQUS) | $M_{cr}$ (Equation (3)) | $M_{cr}$ (GBTUL) | $M_e$ (CUFSM) | $M_{cr}$ (ABAQUS) |
| | kN.m | kN.m | kN.m | kN.m | kN.m | kN.m | kN.m | kN.m |
| Ψ = 1.0 | 222.23 | 2.41 | 2.25 | 2.41 | 55.56 | 0.86 | 0.81 | 0.86 |
| Ψ = 0.5 | 291.13 | 3.18 | 2.81 | 3.19 | 73.21 | 1.13 | 1.01 | 1.13 |
| Ψ = 0.0 | 393.35 | 4.47 | 3.75 | 4.49 | 98.34 | 1.56 | 1.35 | 1.57 |

**Table 6.** Results of the uplift critical moments for the 300 × 2.5 profile.

| Loading Cases | Profile 300 × 2.5 (*L* = 4.0 m) | | | | Profile 300 × 2.5 (*L* = 9.0 m) | | | |
|---|---|---|---|---|---|---|---|---|
| | $M_{cr}$ (Equation (3)) | $M_{cr}$ (GBTUL) | $M_e$ (CUFSM) | $M_{cr}$ (ABAQUS) | $M_{cr}$ (Equation (3)) | $M_{cr}$ (GBTUL) | $M_e$ (CUFSM) | $M_{cr}$ (ABAQUS) |
| | kN.m | kN.m | kN.m | kN.m | kN.m | kN.m | kN.m | kN.m |
| Ψ = 1.0 | 576.30 | 4.56 | 4.44 | 3.51 | 113.84 | 1.41 | 1.37 | 0.97 |
| Ψ = 0.5 | 754.95 | 6.01 | 5.55 | 4.65 | 149.85 | 1.86 | 1.72 | 1.28 |
| Ψ = 0.0 | 1020.05 | 8.40 | 7.40 | 6.54 | 201.49 | 2.55 | 2.29 | 1.78 |

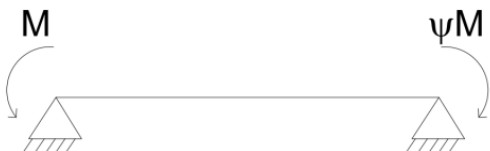

**Figure 20.** Scheme of the loading pattern.

The analysis of Tables 4–6 led to the following observations:

- The critical moment values obtained with Equation (3) are significantly overestimated when compared to those derived from numerical analyses using GBTUL, ABAQUS, and CUFSM. This overestimation is more pronounced for smaller spans.
- When comparing the results obtained with GBTUL, ABAQUS, and CUFSM, a good correlation between the three numerical approaches is observed, with ABAQUS generally providing lower values of critical moments. An exception is noted for the intermediate profile (170 × 2.0), where CUFSM leads to slightly lower values than ABAQUS and GBTUL. Nevertheless, for this profile, the values of ABAQUS and GBTUL are practically coincidental.

Tables 7 and 8 present the bending moment resistance to global buckling obtained for the 70 × 1.5 and 170 × 2.0 profiles, computed according to the European [2], Brazilian [6], and North American [4] standards. The values are based on the critical moments obtained for the higher span lengths (*L* = 4.0 m and *L* = 6.0 m). It is noteworthy that the results for the

$300 \times 2.5$ profiles are not presented due to the normalized slenderness values significantly exceeding the limits set by the standards.

**Table 7.** Resistant bending moment to global buckling for the $70 \times 1.5$ profile.

| | Profile $70 \times 1.5$ ($L = 4.0$ m) | | | | | |
|---|---|---|---|---|---|---|
| | $M_{b,Rd}$ [2] (kN.m) | | | $M_{Rd}$ [4,6] (kN.m) | | |
| | (Equation (3)) | (GBTUL) | (CUFSM) | (ABAQUS) | (GBTUL) | (CUFSM) | (ABAQUS) |
| $\Psi = 1.0$ | 2.01 | 0.22 | 0.23 | 0.22 | 0.23 | 0.26 | 0.22 |
| $\Psi = 0.5$ | 2.13 | 0.29 | 0.29 | 0.28 | 0.30 | 0.29 | 0.29 |
| $\Psi = 0.0$ | 2.21 | 0.38 | 0.37 | 0.37 | 0.40 | 0.39 | 0.39 |

**Table 8.** Resistant bending moment to global buckling for the $170 \times 2.0$ profile.

| | Profile $170 \times 2.0$ ($L = 6.0$ m) | | | | | |
|---|---|---|---|---|---|---|
| | $M_{b,Rd}$ [2] (kN.m) | | | $M_{Rd}$ [4,6] (kN.m) | | |
| | (Equation (3)) | (GBTUL) | (CUFSM) | (ABAQUS) | (GBTUL) | (CUFSM) | (ABAQUS) |
| $\Psi = 1.0$ | 12.13 | 0.78 | 0.74 | 0.78 | 0.78 | 0.74 | 0.78 |
| $\Psi = 0.5$ | 12.51 | 1.01 | 0.91 | 1.01 | 1.03 | 0.92 | 1.03 |
| $\Psi = 0.0$ | 12.84 | 1.38 | 1.20 | 1.38 | 1.42 | 1.23 | 1.43 |

The analysis of the tables highlights the dependency of the bending moment resistance on the critical moment value. The variations observed in the critical moment values using different approaches are also reflected in the values of the bending moment resistance to global buckling, regardless of the adopted standard. Nevertheless, the differences are not as pronounced as those observed for the critical moment results. For example, in the case of the $170 \times 2.0$ profile with a 6.0-m span, the critical moment values obtained with Equation (3) are 62 times higher than the critical moment values obtained with ABAQUS. However, for the same profile with the same span, the bending moment resistance, calculated through Eurocode 3, using Equation (3), is only nine times higher than the bending moment resistance obtained using ABAQUS.

Regarding the comparison between standards, no relevant differences were found, particularly when the GBTUL, CUFSM, and ABAQUS software were used.

## 6. Conclusions

The objective of this study was to assess the bending moment resistance to the global buckling of Cold-Formed Steel (CFS) omega-shaped profiles. Current design codes emphasize the importance of the elastic critical lateral–torsional buckling moment in evaluating the resistance to global buckling, ensuring the safety of steel members. To investigate this parameter's impact, we conducted a comparative analysis involving numerical simulations with various computational tools, namely ABAQUS, GBTUL, and CUFSM, as well as the analytical expression proposed in ENV 1993-1-1:1992 [19]. Our study focused on a selection of cold-formed omega-shaped profiles, chosen based on their height and thickness.

When assessing the sectional bending moment resistances, we observed consistency among the European, Brazilian, and American standards. However, variations emerged when evaluating the bending moment resistance to global buckling. These discrepancies were linked to differences in assessing the critical lateral–torsional moment through various approaches, albeit to a lesser extent. Specifically, the bending moment resistance calculated according to Eurocode 3, using an analytical expression proposed in the past for hot-rolled profiles, exhibited a significant overestimation compared to numerical methods.

Therefore, we draw a key conclusion regarding the inadequacy of using the standard analytical equation for evaluating the critical lateral–torsional buckling of laminated steel beams when predicting the critical moment of CFS omega-shaped profiles. The agreement

between GBTUL, CUFSM, and ABAQUS critical moments suggests their reliability for estimating the elastic critical lateral–torsional buckling moment. Certainly, GBTUL stands out for its simplicity, ease of application, and capability to incorporate various types of loading, rendering it a favorable option for evaluating this parameter.

It is important to note that this study did not account for the beneficial effect of cladding sheets. These sheets are typically positioned beneath metal purlins to provide points of lateral restraint. Consequently, the bending moment resistance calculated using GBTUL, CUFSM, and ABAQUS may underestimate, in some cases, the capacity of CFS omega-shaped beams. To gain a comprehensive understanding, further experimental studies should consider more realistic loading and boundary conditions.

**Author Contributions:** Conceptualization, L.M., J.L.R.P. and J.M.C.; Methodology, L.M., J.L.R.P. and J.M.C.; Software, J.C. and J.A.E.; Formal analysis, R.P., J.C. and J.A.E.; Writing—original draft, R.P., J.C. and J.A.E.; Writing—review & editing, J.L.R.P. and J.M.C.; Supervision, J.L.R.P. and J.M.C. All authors have read and agreed to the published version of the manuscript.

**Funding:** This work was partially funded by the Agenda "R2U Technologies—modular systems", contract C644876810-00000019, investment project 48, funded by the Recovery and Resilience Plan (PRR) and the European Union—Next Generation EU. This research was also supported by UID/ECI/04708/2019—CONSTRUCT—Instituto de I&D em Estruturas e Construções, funded by national funds through the FCT/MCTES (PIDDAC).

**Data Availability Statement:** Data are contained within the article.

**Acknowledgments:** The authors would like to thank Professor Rodrigo Gonçalves from NOVA University Lisbon for the support provided for the use of the GBTUL software.

**Conflicts of Interest:** The authors declare no conflicts of interest.

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
