# Peer review of "Global Buckling Resistance of Cold-Formed Steel Beams with Omega-Shaped Sections"

_applsci, doi:10.3390/app14093857_

Round 1

Reviewer 1 Report

Comments and Suggestions for Authors

A comparative study was conducted to explore various approaches for calculating the critical moment. Using different methods available in codes and computational tools. The analysis followed the Effective Width Method, employing the expression proposed in ENV 1993-1-1:1992, which is commonly used for evaluating the critical lateral-torsional moment of hot-rolled profiles. The ABAQUS model, validated with results obtained from an experimental campaign, serves as the reference model. Upon assessing the bending moment resistances according to European, Brazilian, and American standards, consistency was found among these standards. It became evident that using the analytical expression proposed for hot-rolled profiles is inadequate for evaluating the critical lateral-torsional moment of CFS Omega-shaped profiles. The agreement between CUFSM and ABAQUS results suggests their utility as reliable tools for estimating the elastic critical lateral-torsional buckling moment. It has reference value for evaluating the critical moment of lateral torsional buckling of cold-formed beams with Omega shaped cross-sections.

Reviewer 2 Report

Comments and Suggestions for Authors

The purpose of this study was to assess the bending moment resistance to global buckling of CFS omega-shaped profiles. A comparative analysis was made between Eurocode, Brazilian and American standards.

The authors conducted experiments on the such cold-formed profiles and validated them using numerical approaches such as CUFSM and ABAQUS. They have suggested the CUFSM approach to be more reliable in estimating the lateral-torsional buckling moment. 

The study, according to the authors, is the first of its kind. The key finding of this research is that the study has found the standard equation for calculating the critical lateral torsional buckling to be inadequate. 

In lines 305 through 309, the authors state that the EWM formulation give slightly lower values than the test values indicating conservative estimates for 70x1.5 and 170x2 profiles. However, the comparison shows that the results for 170x2 shows significantly high load values (almost 6 times). DSM formulation, however, provides reasonable estimates. The authors are invited to comment on this.

Additionally, the purpose of experimental study was to find the efficacy of the EWM and DSM methods, the numerical modeling was simply carried out to validate the experimental work. Although the authors have found some key points in the numerical results, this modeling becomes redunent in the absence of any parametric study. 

Furthermore, it is imperative to make a theoretical comparison between the numerical approaches and the key differences be pointed out rather than making a distinction between software packages.

Based on the review of article, the authors have provided a detailed explanation of different codes but have not successfully established what advantages omega-shaped profiles provide compared to the other profiles which could emphasize the importance of looking into the design  of these profiles. For starters, a comparison of the shape advantage to some common shapes based on materials and cross-sectional properties could be helpful.

The experiment design is thorough and gives insights into the failure but the numerical modeling seems redundant when the major significance is the approach comparison and experimental validation. A parametric study could have given more insights into the behavior.

Based on the experimental and numerical results, the authors have pointed out some key flaws in the buckling resistance calculation formulations. Perhaps, the capabilities of numerical study could have provided additional insights into improving the existing formulations.

The authors are invited to establish the significance of omega-shaped profiles. Additionally, if there is a computational difference in solving for different software packages, it should be highlighted. Lastly, there is no significant contribution of the simulation in the solution. The result of these three issues is the lack of novelty and must be addressed in order to be published in the journal.

Comments on the Quality of English Language

There may be one instance where I detected grammatical error (line 483-484). Rest is all perfectly fine.

Reviewer 3 Report

Comments and Suggestions for Authors

This paper aims to study the response of cold formed steel beams with omega-shaped section under lateral loading. Different methods including analytical and numerical approaches have been used. The paper needs major revision before publishing and the comments are as follows:

1- The introduction part should be further enriched with the new published papers in this area, since most of the used papers belong to before 2022.

2- In section 3.1, the lengths of tested profiles should be provided.

3- Further details should be given in section 3.2 about the applied load rate, the way of acquiring data, details about the supports, etc.

4- Why upward load should be applied? How is the upward load applied to the specimen and how is the support condition provided in this phase of loading? Provide more explanations.

5- In fig. 3a, three curves are shown as Test 1 to Test 3, but in the previous section and also in Table 3, only two sections are considered for this section properties. Please clarify.

6- The description of Fig. 3 to 5 should be further extended with more details.

7- What values are considered for modulus of elasticity and density of the materials?

8- In the context, reference 18 is not given, while the authors use references 19 and 20. Furthermore, references 18-20 are not provided in the reference part of the paper.

9- What are the material properties and dimensions of other parts of the test setup that have been modeled in Abaqus? Provide more details.

10- Why is the friction coefficient 0.35 used? Provide a reference.

11- How is the examined profile constrained against upward load during the experiment? How is the boundary condition provided? Provide more details.

12- Provide more explanation about Fig. 12 to show there is consistency and agreement between numerical and experimental results. Also, add the error percentage between numerical and experimental results.

13- Provide an explanation about the modeling with GBTUL and CUFSM. Just imagine that a reader of the paper wants to learn these methods as much as possible from your paper. Without any details or introduction about them, the results are provided and compared with Abaqus. Therefore, in the revised manuscript, a detailed explanation should be provided.

14- Why are the results of Abaqus considered as a reference? It is better if the results of all methods are compared with experimental results, since the Abaqus results may also have some errors with experimental results.

15- In the conclusion part, final part, some results are given for cladding sheets, which are for the first time discussed in the conclusion part. If it is related to the objective of the current study, it should be discussed in the context when the other results are provided in terms of experimental and numerical ones. In other words, this should be explained first in the paper, and then concluded in the conclusion part.

Comments on the Quality of English Language

Minor editing of English language required.

Round 2

Reviewer 2 Report

Comments and Suggestions for Authors

The authors have responded to the three major issues raised.

The significance of using omega-shape has been attributed to the improvement in flexural capacity and stiffness.

There was a typo in comparison, it has been rectified.

The authors carried out a comparison of several software packages. And parametric study was not required because it was a comparison for quantifying critical moments. 

The points have been answered. 

Reviewer 3 Report

Comments and Suggestions for Authors

All comments are addressed in the revised paper, and it is well improved. I have no further comments for the authors and the paper in its current form is acceptable to publish in this journal.